# Alterations in the Kynurenine–Tryptophan Pathway and Lipid Dysregulation Are Preserved Features of COVID-19 in Hemodialysis

**DOI:** 10.3390/ijms232214089

**Published:** 2022-11-15

**Authors:** Max Schuller, Monika Oberhuber, Barbara Prietl, Elmar Zügner, Eva-Maria Prugger, Christoph Magnes, Alexander H. Kirsch, Sabine Schmaldienst, Thomas Pieber, Marianne Brodmann, Alexander R. Rosenkranz, Philipp Eller, Kathrin Eller

**Affiliations:** 1Division of Nephrology, Department of Internal Medicine, Medical University of Graz, 8036 Graz, Austria; 2Center for Biomarker Research in Medicine, CBmed GmbH, 8010 Graz, Austria; 3Division of Diabetes and Endocrinology, Department of Internal Medicine, Medical University of Graz, 8036 Graz, Austria; 4Institute for Biomedicine and Health Sciences (HEALTH), Joanneum Research Forschungsgesellschaft m.b.H., 8010 Graz, Austria; 5Klinikum Favoriten, Wiener Krankenanstaltenverbund, 1100 Vienna, Austria; 6Division of Angiology, Department of Internal Medicine, Medical University of Graz, 8036 Graz, Austria; 7Intensive Care Unit, Department of Internal Medicine, Medical University of Graz, 8036 Graz, Austria

**Keywords:** COVID-19, metabolomics, hemodialysis patients, kynurenine, lipid dysregulation

## Abstract

Coronavirus disease 2019 (COVID-19)-induced metabolic alterations have been proposed as a source for prognostic biomarkers and may harbor potential for therapeutic exploitation. However, the metabolic impact of COVID-19 in hemodialysis (HD), a setting of profound a priori alterations, remains unstudied. To evaluate potential COVID-19 biomarkers in end-stage kidney disease (CKD G5), we analyzed the plasma metabolites in different COVID-19 stages in patients with or without HD. We recruited 18 and 9 asymptomatic and mild, 11 and 11 moderate, 2 and 13 severely affected, and 10 and 6 uninfected HD and non-HD patients, respectively. Plasma samples were taken at the time of diagnosis and/or upon admission to the hospital and analyzed by targeted metabolomics and cytokine/chemokine profiling. Targeted metabolomics confirmed stage-dependent alterations of the metabolome in non-HD patients with COVID-19, which were less pronounced in HD patients. Elevated kynurenine levels and lipid dysregulation, shown by an increase in circulating free fatty acids and a decrease in lysophospholipids, could distinguish patients with moderate COVID-19 from non-infected individuals in both groups. Kynurenine and lipid alterations were also associated with ICAM-1 and IL-15 levels in HD and non-HD patients. Our findings support the kynurenine pathway and plasma lipids as universal biomarkers of moderate and severe COVID-19 independent of kidney function.

## 1. Introduction

Since the outbreak of the coronavirus disease 2019 (COVID-19) pandemic, at least 601 million people have been infected and 6.4 million have died from the disease as of September 2022 [1]. Symptoms of severe acute respiratory syndrome coronavirus 2 (SARS-CoV-2) infection range from oligo/-asymptomatic presentations to severe acute-respiratory distress syndrome [2]. Despite significant advances in COVID-19 treatment and vaccination (reviewed recently [3,4]) and in light of novel variants [5], timely identification of individuals at high risk for severe and fatal courses remains of great clinical importance.

Accordingly, the COVID-19 metabolome has been investigated in search of prognostic biomarkers and potential therapeutic targets. Metabolic profiling of bodily fluids, particularly plasma or serum, either by mass spectrometry or nuclear magnetic resonance, is attractive for several reasons: First, sampling is usually minimally invasive (i.e., drawing of peripheral blood). Second, sample workup is relatively easy compared to next-generation sequencing or proteomic analyses. Third, the metabolome provides valuable insight in the net metabolic state.

COVID-19 leads to profound stage-dependent metabolic alterations [6,7,8,9,10], and disturbances in the kynurenine pathway have been repeatedly reported in severe COVID-19 [6,7,8,10]. Kynurenine (Kyn) is derived from tryptophan (Trp) via the kynurenine pathway, and this reaction is catalyzed by indole 2,3-dioxygenase 1 (IDO). IDO, in turn, is activated under inflammatory conditions [11]. IDO induction results in an increase in the Kyn:Trp ratio, which leads to the production of anti-inflammatory mediators like interleukin (IL-) 10 and promotes differentiation of T-cells into regulatory T-cells [12]. Kyn and other metabolites of the Kyn pathway have been proposed as potential biomarkers for COVID-19 progression [6,7,8]. Furthermore, IDO inhibitors may enhance antiviral activity in COVID-19 [8].

Lipid dysregulation is another hallmark of moderate and severe COVID-19 with potential prognostic and therapeutic implications [7,13,14]. SARS-CoV-2 can reprogram host metabolism towards de novo lipogenesis to promote rapid viral replication [15]. Furthermore, secreted phospholipase A_2_ type IIA (sPLA_2_ IIA) activity increases with disease severity [16]. sPLA_2_ IIA cleaves membrane phospholipids into lysophospholipids (LPL) and free fatty acids (FFA). Excessive release of these lipid mediators exacerbates the inflammatory response. Therefore, sPLA_2_ IIA inhibitors have been proposed as therapeutic avenue in COVID-19.

Advanced chronic kidney disease (CKD) and hemodialysis (HD) have been identified as major risk factors for severe COVID-19 [17], although this has been challenged by others [18]. Loss of kidney function results in an “uremic state” [19], and in the disarray of small molecules in HD patients compared to non-CKD individuals [20]. In concert with profound immunological alterations [21], these factors create a different playing field for infections in general and COVID-19 specifically. Hence, extrapolation of metabolomic data from non-CKD patients may be invalid. Moreover, we have previously shown that the immune response to COVID-19 is dampened in HD patients compared to non-HD individuals, potentially due to an overactivated inflammatory state at baseline [22].

We analyzed the circulating metabolome of HD and non-HD individuals with different stages of COVID-19 to evaluate whether COVID-19-induced changes in metabolite levels are reproducible in HD patients. Furthermore, we aimed to evaluate whether the metabolome provides clues on the inflammatory state with the potential for therapeutic exploitation.

## 2. Results

### 2.1. Hemodialysis Impacts Profoundly on the Circulating Metabolome Irrespective of COVID-19 Status

Eighty individuals were included prospectively in our study [22]. Individuals were grouped according to COVID-19 status and severity and further stratified according to HD status into the following groups: non-HD negative (*n* = 6), HD negative (*n* = 10), non-HD asymptomatic/mild (*n* = 9), HD asymptomatic/mild (*n* = 18), non-HD moderate (*n* = 11), HD moderate (*n* = 11), non-HD severe (*n* = 13), and HD severe (*n* = 2) (Figure 1A). Severe cases were in need of intensive care, i.e., admitted to an intensive care unit, died prematurely, or rejected due to triage decision. Moderate cases were hospitalized, but without need for intensive care. Individuals with an asymptomatic/mild disease course could be managed without hospitalization. “Negative” individuals were included as a COVID-19 negative control group for HD and non-HD, respectively. Baseline characteristics have been previously published [22]. Principal component (PC) analysis of circulating metabolites revealed a substantial impact of HD at baseline (Figure 1B). This separation along the PC-2 axis of HD and non-HD patients was maintained in COVID-19 and throughout its clinical evolution. A rightwards shift on the PC-1 dependent axis was observed in non-HD patients with increasing COVID-19 severity. A similar, albeit smaller, tendency could be seen in HD patients (Figure 1B). As expected, PC-2 was influenced by markers of uremia and HD-associated metabolic alterations [20] (Table 1). PC-1 was composed mostly of mono- or polyunsaturated free fatty acids (FFAs) (Table 1). The changes in circulating FFAs, paralleled by LPL alterations with increasing severity, were evident in HD and non-HD patients and hint towards disturbances in fatty acid metabolism in COVID-19 (Figure 2).

### 2.2. COVID-19-Dependent Metabolic Alterations Are Less Pronounced in Hemodialysis

Non-HD patients displayed a COVID-19 severity-associated metabolic signature (Figure 1B and Figure 2). A similar effect was observed in HD patients, though COVID-19 dependent alterations were less pronounced than in non-HD individuals (Figure 1B and Figure 2). For non-HD, asymptomatic/mild patients did not differ from COVID-19 negative individuals, but 61 and 70 differentially expressed metabolites were found in the comparison between negative and moderate and severe patients, respectively (Figure 1C and Figure 3). In the HD stratum, only two metabolites were significantly deregulated between negative and asymptomatic/mild patients (Figure 1C), whereas 15 metabolites were altered between negative and moderate cases (Figure 1C and Figure 3). Due to the small number of severe cases in the HD population, we were not able to compare this group to others. However, as indicated by the large overlap of deregulated metabolites in non-HD negative vs. moderate and non-HD negative vs. severe contrasts (Figure 3), the metabolome appears to be critically altered already in moderate disease.

### 2.3. Preserved Metabolite Deregulations in Moderate COVID-19 in Hemodialysis Patients

Nine metabolites showed overlapping changes in HD and non-HD patients with moderate COVID-19 compared to negative individuals (Figure 4A). Boxplots of these metabolites through all clinical stages are depicted in Figure 4B. Whereas proline (Appendix A), lysophosphatidylethanolamine (lysoPE) (18:1), lysoPE(18:2), lysophosphatidylcholine (lysoPC) (18:2), FFA C24:1, FFA C24:2, FFA C22:2, and xanthine (Appendix A) showed similar dynamics in non-HD and HD patients with increasing disease severity, two important observations in Kyn regulation ought to be noted: First, Kyn was the only metabolite to differentiate between moderate and severe COVID-19 in non-HD patients. Second, Kyn levels in HD patients without COVID-19 were already comparable to non-HD patients with moderate disease. Moreover, the Kyn:Trp ratio was calculated, as a rough estimation of IDO activity [23]. Mirroring Kyn levels, we found an elevated Kyn:Trp ratio in HD patients and an increase with COVID-19 severity independent of HD status (Figure 4C). These findings suggest that Kyn and the Kyn:Trp ratio are regulated by both severe COVID-19 and CKD G5. While Kyn and the Kyn:Trp ratio increased in both HD and non-HD, Trp levels declined only in non-HD patients with increasing disease severity. Trp abundance remained at low but stable levels in COVID-19 cases on HD (Figure 4D).

Boxplots of metabolites, which were uniquely regulated in HD patients with COVID-19, are depicted in Appendix A. Arginine and glyceric acid showed a significant decline, whereas taurine and niacinamide were upregulated with disease severity only in HD patients. FFA C10:0 and FFA C11:0 were markedly upregulated in HD patients already at the asymptomatic/mild stage of disease evolution without further dynamics at higher stages. Of note, FFA C10:0 and FFA C11:0 were the only two significantly deregulated metabolites between negative HD patients and HD patients with asymptomatic/mild COVID-19 (Figure 1C).

### 2.4. Kynurenine and Kynurenine:Tryptophane Ratio Correlate with Creatinine in Non-Hemodialysis Patients with COVID-19

We observed that some non-HD patients with moderate or severe disease have increased creatinine levels compared to controls (Appendix A), which may indicate acute kidney injury [24] or pre-existing CKD. Advanced CKD stages (G3b and G4) were rare with a frequency of 0% (0/6), 11.1% (1/9), 9.1% (1/11), and 30.8% (4/13) for negative, asymptomatic/mild, moderate, and severe cases, respectively [22]. However, we observed increased creatinine levels in 18.1% (2/11) moderate and 61.5% (8/13) severe cases, indicating acute kidney injury in those patients (Appendix A). To evaluate whether shared metabolites between HD and non-HD (Figure 4B) are linked with creatinine abundance, we performed correlational analysis. In contrast to the other metabolites, Kyn and the Kyn:Trp ratio showed a moderate and positive correlation with creatinine (Appendix A).

### 2.5. Integration of Cytokine and Chemokine Data Links Metabolites with Inflammatory Response to COVID-19

Immune response to COVID-19 impacts on circulating metabolites and vice versa [8]. Furthermore, evidence suggests that the inflammatory response to COVID-19 is critically altered in individuals requiring HD [22]. We selected the “shared metabolites” (Figure 4B) and correlated their abundance to plasma cytokine and chemokine concentrations for non-HD and HD patients separately (Figure 5A,B). Kyn levels were positively correlated with intercellular adhesion molecule 1 (ICAM-1) and IL-15 in both groups, and with vascular cell adhesion molecule 1 (VCAM-1) and IL-6 in non-HD patients only (Figure 5A,B and Figure 6). Furthermore, Kyn showed a positive association with IL-10 in HD patients (Figure 5B and Figure 6). Similar correlations were observed with the Kyn:Trp ratio (Figure 5A,B and Figure 6). Interestingly, in the HD population Kyn:Trp ratio paralleled the expression of key inflammatory mediators, interferone gamma (IFNγ), and tumor necrosis factor alpha (TNFα) (Figure 5B). Additionally, lysophospholipids (lysoPE(18:1), lysoPE(18:2), and lysoPC(18:2)) and proline were all inversely correlated with IFNγ, IL-15, and IL-6 in the HD population, whereas in non-HD patients, only lysoPE(18:2) and lysoPC(18:2) were negatively associated with IL-15 (Figure 5A,B). In neither stratum, CRP was significantly correlated with any metabolite (Figure 5A,B). Thus, despite an attenuated inflammatory response to COVID-19 [22], metabolic alterations, especially Kyn abundance and the Kyn:Trp ratio, are intertwined with cytokine/chemokine release in HD. This interplay is partly overlapping with non-HD patients.

## 3. Discussion

In the present study, we demonstrate a substantial impact of CKD G5 on the circulating metabolome, which is maintained throughout all clinical stages of COVID-19. In accordance with the existing literature, we could show that there are COVID-19 stage-dependent alterations in non-HD patients. To our knowledge, we are the first to report that metabolic changes in HD patients with COVID-19 are less pronounced but still overlap with the response in non-HD individuals. Importantly, lipid dysregulation, i.e., elevated circulating FFAs and diminished LPLs, and increases in Kyn and the Kyn:Trp ratio, appear to be conserved in HD. We further provide observational evidence of immune–metabolic interplay, supporting the notion that metabolism modulates inflammation. The observed changes in circulating metabolites due to COVID-19 are of great clinical importance for the HD-population as metabolome-derived prognostic markers, and possible therapeutic targets in COVID-19 may be critically altered.

Loss of kidney function and its sequelae profoundly perturb the circulating metabolome [20,25]. HD allows for the removal of uremic toxins, but it is non-continuous, and dializability of compounds may vary greatly [26]. Thus a metabolic signature similar to kidney healthy individuals cannot be achieved by HD [25]. The differences between HD negative and non-HD negative individuals reflect this “uremic state”. Thus, the influence of CKD G5 was not masked by COVID-19, underlining its major impact on the net metabolic state.

We observed increased creatinine levels in some non-HD patients with moderate and severe COVID-19. Importantly, these could not be fully explained by advanced CKD stages (G3b and G4) at baseline, as these were rare in our non-HD cohort. Increased creatinine at the time of blood sampling may thus indicate acute kidney injury (AKI), though we recognize the limitations of creatinine as markers of AKI in critical illness [27]. AKI is a common COVID-19 related complication with severe implications [28,29] and has been shown to influence circulating metabolites [30]. Although our study was neither designed nor powered to assess the impact of AKI on plasma metabolites, we observed a positive association of Kyn and the Kyn:Trp ratio with creatinine in non-HD patients. However, only half of moderate and severe cases had increased creatinine levels, and these were only slightly increased (as compared to HD). Furthermore, creatinine levels alone were not indicative of a moderate or severe course in non-HD. In contrast, Kyn and the Kyn:Trp ratio were robustly and almost uniformly elevated in moderate and severe disease, hence providing valuable information on disease severity independently of kidney function.

Inflammatory response and microbial defensive mechanisms are negatively affected by uremia [31]. Register analyses have reported an excess mortality of CKD G5 patients with COVID-19 [32], although it is difficult to abstract the influence of CKD G5 without a comparator group with similar risk factors. When propensity matched for age and comorbidities, Chan et al. found no difference in in-hospital mortality between patients with kidney failure and those without kidney failure [18]. Likewise, the number of patients with severe COVID-19 and overall mortality was lower in the HD group of our study population [22]. We proposed that chronic low grade inflammation, as evident from blood leucocytes and circulating cytokines and chemokines, may protect HD patients from COVID-19 induced hyperinflammation and its deleterious consequences [22]. The present study adds that the dampened immune response is paralleled by a dampened metabolic response to COVID-19.

The COVID-19 metabolome in non-HD patients has been extensively studied since the outbreak of the pandemic and alterations in the circulating metabolite profile have been well characterized. Thomas et al. were the first to describe a role for the Kyn pathway in COVID-19 [7], and others have confirmed Kyn and associated metabolites as biomarkers of severe and fatal infection [6,8,10]. Presently, Kyn expression increased with COVID-19 severity in non-HD and HD patients, although the baseline levels differed significantly. Specifically, COVID-19-negative individuals on HD showed similar Kyn abundance as non-HD patients with moderate COVID-19. Pawlak et al. have shown that Kyn and its metabolites accumulate with declining kidney function [33]. IDO activity reportedly increases with CKD stage [34], indicating that the rise in Kyn levels in CKD is explained, at least partially, due to an increase in Trp metabolization rather than reduced filtration only [35]. Pro-inflammatory cytokines, such as TNFα and interferons, can upregulate IDO [36]. Thus, enhanced IDO activity, as seen in CKD [35] and COVID-19 [37], may result from increased inflammatory signaling. Furthermore, the higher baseline levels of Kyn may be a consequence of chronic inflammation in uremia. In spite of potential chronic activation in HD, our correlational evidence suggests an intact IFNγ/TNFα-IDO-Kyn axis sensitive to COVID-19. Taken together, these results support a different “normal range” for Kyn in CKD stage 5. Plasma Kyn levels increase with COVID-19 severity, but the prognostic role of Kyn in HD patients remains to be clarified.

In agreement with others [7], higher IL-6 levels, a prognostic marker in COVID-19 [38], coincided with a surge in Kyn abundance in non-HD patients. The lack of such a connection in HD may be explained by the observation that IL-6 peaks in severely affected individuals [39] and by the low number of critically ill patients in the HD stratum. Furthermore, owing to the low-grade inflammation, IL-6 has been found to be chronically elevated in uremia [40].

Clearance of viral pathogens by NK cells and CD8^+^ T cells is mediated by IL-15 [41]. Recently, CAR NK cells, engineered to express soluble IL-15 to promote their survival, have shown promising results in COVID-19 therapy [42], underlining the potential of IL-15 as viable treatment option. Furthermore, IL-15 signaling may be paramount in the humoral response to infection and vaccination [43]. Here, IL-15 was positively correlated with Kyn independently of the HD status. Of note, cytotoxic effector functions of NK cells have been suggested to be dependent on IDO [44]. To our knowledge, a direct link between IDO activity and IL-15 is not known, but it is tempting to speculate, that both, IL-15 and Kyn (derived via IDO), may promote cytotoxicity in viral infections.

Effects of Kyn and the Kyn pathway are directed towards immune suppression via various mechanisms [45]. Accordingly, macrophage IL-10 production has been shown to increase in an IDO-dependent manner [46]. Danlos et al. have shown that IL-10 levels correlate with anthranilic acid, a metabolite of Kyn, in COVID-19 patients [6]. In our hands, Kyn levels correlated with IL-10 abundance in HD but not in non-HD patients. Generally regarded as an anti-inflammatory cytokine, dramatically elevated IL-10 levels are found in severely affected COVID-19 patients together with pro-inflammatory mediators, a unique feature of COVID-19 related cytokine storm [47]. Furthermore, IL-10 correlates with poor survival [48]. Possible explanations for this paradox include IL-10 resistance and/or non-traditional pro-inflammatory functions [49].

ICAM-1 and VCAM-1, soluble markers of endothelial inflammatory activation [50], rise in severe COVID-19 cases [51]. As they are critical for leucocyte recruitment, higher circulating levels may reflect more severe (lung) inflammation. Kyn expression was correlated with ICAM-1 in both HD and non-HD patients, and with VCAM-1 only in non-HD patients. These findings echo results from an observational study in CKD patients with atherosclerosis, where Kyn was independently associated with ICAM-1, whereas VCAM-1 was linked to oxidative status and platelets [35].

Another anti-inflammatory mechanism of IDO activation is Trp deprivation, which may hinder T-cell activity [52]. Consistently with previous reports [7,8], Trp was negatively regulated in our non-HD population with COVID-19. Trp levels gradually decline in CKD and are lowest in patients on HD [53]. The exact nature of Trp deficiency and turnover in HD remain incompletely understood [54], though Schefold et al. highlighted IDO-dependent Trp metabolization as a major contributor [34]. Presently, HD patients with COVID-19 displayed low Trp levels, which were stable throughout all clinical stages, despite an increased Kyn:Trp ratio as surrogate for IDO activation. Further research is necessary to explore how Trp levels can be maintained in HD patients in this setting and if Trp supplementation could aid in viral clearance. However, excess dietary Trp may result in indoxyl sulphate accumulation, a potentially toxic uremic metabolite that cannot be cleared by HD [55].

Ultimately, metabolization of Trp through the Kyn pathway generates nicotinamide adenine dinucleotide (NAD^+^), a vital cofactor in mitochondrial oxidation and fuel for enzymes with antiviral activity [56,57]. Apart from the IDO-dependent de novo generation of NAD^+^, nicotinamide (also known as niacinamide) may be converted to NAD^+^ via the intermediate nicotinamide mononucleotide through a “salvage pathway” [57]. Evidently SARS-CoV-2 can deplete cellular NAD^+^ [58]. We observed increased levels of circulating nicotinamide in HD patients with COVID-19, which may indicate an upregulation of key enzymes of the salvage pathway in an attempt to restore NAD^+^ levels [58]. Measuring NAD^+^ levels and enzymatic activity in PBMCs of patients on HD with COVID-19 could provide further insight. Interestingly, nicotinamide levels remained stable even in severely affected non-HD patients, even though disturbances of the NAD^+^ pathway have been described by others and were correlated with cytokine release [8]. Notably, NAD^+^ boosters such as nicotinamide or nicotinamide riboside have improved outcomes in acute COVID-19 related kidney injury [59], and (in conjunction with other cofactors) recovery time and circulating cytokines [60], respectively.

Diminished NO bioavailability due to an altered arginine proline metabolism is a major contributor to endothelial dysfunction and a hallmark of severe COVID-19 [61]. Our results confirm these previous reports [8,62] and add that similar changes of proline and arginine can be seen in HD patients with moderate disease. Furthermore, reduced proline levels were associated with increasing abundance of IFNγ, IL-15, and IL-6 in this population. Intriguingly, the arginine proline metabolism appears to be a preserved feature of moderate/severe COVID-19 in HD, despite an already impaired vascular endothelium in uremia [63].

In line with others [7,13,14], we could show a broad derangement of lipid metabolism including FFAs and lysoPE isoforms in COVID-19 regardless of HD status, supporting roles as universal biomarkers of COVID-19 severity for these compounds. These findings further suggest underlying mechanisms that are conserved in HD [13]. In accordance with Sindelar et al. [13], we showed that plasma lipid alterations are tightly linked to SARS-CoV-2 related hyperinflammation. Thus, providing evidence for immune-metabolic crosslinking in HD patients. Excessive hydrolization of cellular and mitochondrial membranes due to increased plasma sPLA_2_ IIA activity, as seen in COVID-19, could explain evident signs of lipid dysregulation in plasma [16]. This may aggravate tissue damage and inflammatory response, which in turn induces sPLA_2_ IIA expression [64], thereby creating a *circulus vitiosus* [16]. Furthermore, lipid droplet biogenesis and intracellular lipid accumulation may be paramount in viral replication [15], and blunt type-I interferon signaling simultaneously [65]. Our data indicate a need for a closer investigation of COVID-19-induced lipid dysregulation in the HD population, as similar mechanisms may be at play and thus similar treatments, such as inhibition of sPLA_2_ IIA or counteracting lipid droplet formation, may be feasible as in non-HD individuals.

We demonstrated elevated xanthine levels in both groups and an increase in adenine in non-HD patients. These findings hint towards disturbances of the purine metabolism, although other involved metabolites (e.g., adenosine monophosphate and uric acid) were not altered by COVID-19 echoing results from Valdes et al. [66]. Extracellular purines are released in response to inflammation and may modulate the inflammatory response [67]. Moreover, SARS-CoV-2 has been shown to modulate host purine synthesis to support its replication [68]. CKD is commonly associated with hyperuricemia, and experimental studies suggest a potential mechanistic link between purine metabolism and CKD progression [69]. However, neither baseline uric acid nor xanthine levels differed between our HD and non-HD population.

Intriguingly, glyceric acid showed opposing dynamics with increasing COVID-19 severity depending on HD status. Though literature on glyceric acid is scarce, a study in healthy individuals suggests a potential benefit of glyceric acid supplementation with regard to mitochondrial energy utilization and inflammation [70]. However, its HD-dependent peculiar regulation under inflammatory conditions warrants further investigation.

There are certain limitations to our study. First, not all groups were adequately represented. Specifically, the low number of HD patients with severe COVID-19 did not allow for any valid comparisons. On the other hand, this skewed distribution of disease severities in HD patients towards mild/asymptomatic cases may be of relevance in itself [22]. Second, only a single snapshot of the metabolome at the time of diagnosis was taken, without any serial sampling. Third, comorbidities were more frequent in HD patients and may influence circulating metabolites, even though a significant difference was noted only for hypertension [22]. Similarly, medications may have influenced the metabolome. Inherently, targeted metabolomics provides limited coverage of the whole metabolome and does not allow for meaningful pathway analysis. Fourth, the presented study dates back to 2020, and metabolic response may be modified with new SARS-CoV-2 variants and after vaccination. Finally, correlational analysis provides only observational evidence of immune-metabolic interplay, which necessitates experimental confirmation.

## 4. Materials and Methods

### 4.1. Study Population

SARS-CoV-2 positive and negative individuals aged 18 years or older were recruited at the Medical University of Graz and the Hospital Favoriten, Vienna, Austria, as published previously [22]. Briefly, SARS-CoV-2 infection was tested by RT-PCR following a nasopharyngeal swab, and positively tested non-HD individuals and patients on HD were grouped according to disease severity: Asymptomatic and mild cases did not need any medical intervention and could be managed as outpatients. Moderate phenotypes had to be hospitalized. Patients with severe course were in need of intensive care. Furthermore, non-HD negative and HD negative controls without SARS-CoV-2 infection were included. In the non-HD cohort, advanced CKD stages (G3b and G4) were rare with a frequency of 0%, 11.1%, 9.1%, and 30.8% for negative, asymptomatic/mild, moderate, and severe cases, respectively [22]. The baseline characteristics of our study population have been published recently [22].

### 4.2. Metabolomics

Peripheral venous blood was collected in EDTA-plasma for further analysis at time of diagnosis and/or hospital admission. In HD patients, sampling was performed prior to HD. Samples were extracted overnight using cold methanol [71] and measured in a stratified randomized sequence using liquid chromatography-high resolution mass spectrometry. Sample analysis was carried out using an ultra-high performance liquid chromatography Vanquish coupled to a Q-Exactive mass spectrometer (Thermo Fisher Scientific, Waltham, MA, USA) equipped with a NH2-Luna hydrophilic interaction liquid chromatography analytical column as described in [71].

Raw data were converted into mzXML files using msConvert (Proteo Wizard Toolkit v3.0.5, GitHub, San Francisco, CA, USA), and PeakScout (developed by Joanneum Research, Graz, Austria [72]) was used to search for known metabolites by using a reference list containing accurate mass and retention times acquired via reference substances. Metabolite quality control was conducted as described in [73] using TIBCO Spotfire (v7.5.0, TIBCO, Palo Alto, CA, USA). Data were median QC normalized as described previously [74] and afterwards log_10_- transformed.

### 4.3. Cytokine and Chemokine Measurements

Plasma cytokines and chemokines were measured as described previously [22].

### 4.4. Statistical Analysis

All statistical analyses were performed using the R language and environment for statistical computing (version 4.2.2, R Foundation for Statistical Computing, Vienna, Austria). All R packages are listed in Appendix B. Data analysis was performed on median- and quality control (QC)-normalized, log_10_-transformed metabolite data. Principal component analysis and differential expression analysis (limma) were performed. Kyn:Trp ratios were compared by Kruskal–Wallis test and Dunn’s test after testing for normality. Cytokine/Chemokine correlations with metabolites were assessed with Pearson correlation. Benjamini–Hochberg correction was used to correct for multiple testing in all relevant cases. Significance was determined as adjusted *p*-value ≤ 0.05.

## 5. Conclusions

We provide evidence that the Kyn pathway and lipid alterations are preserved biomarkers of COVID-19 severity in the HD population. Meanwhile, Kyn levels are increased in CKD G5 patients per se, calling for adjustment of ranges in CKD patients with COVID-19 in case this biomarker finds its way into the clinical routine. Furthermore, we show an interplay of the immune response and the metabolome providing interesting new therapeutic targets such as IL-15 and ICAM-1.

## Figures and Tables

**Figure 1 ijms-23-14089-f001:**
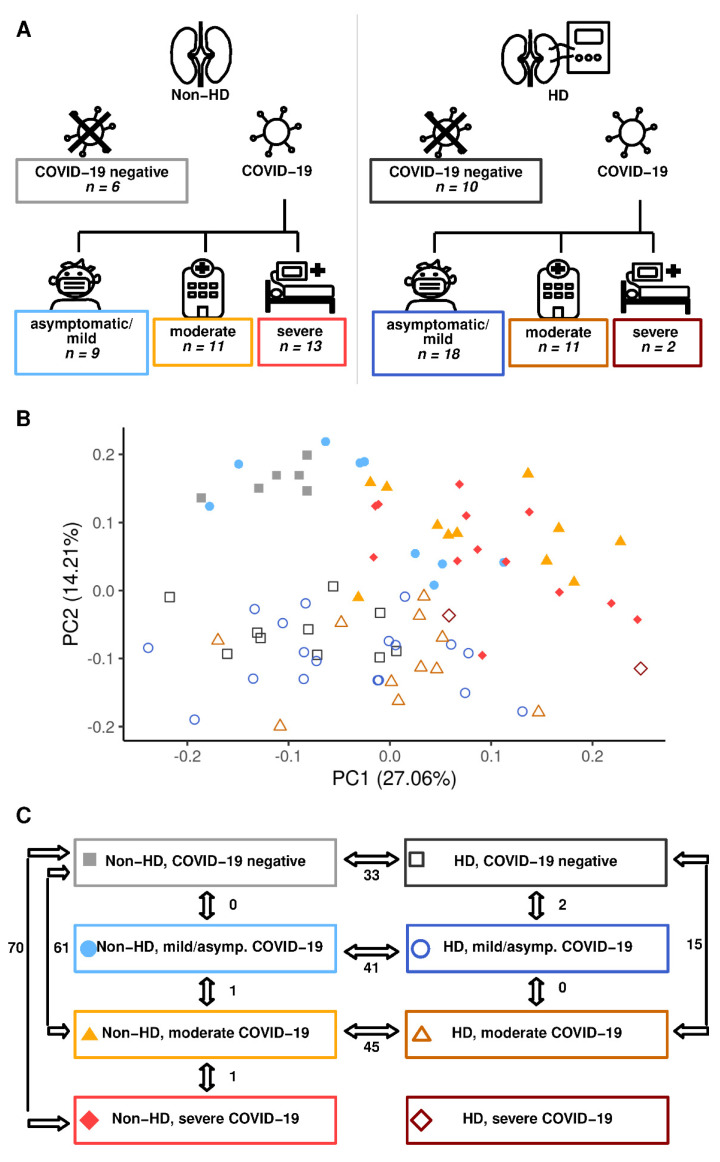
HD profoundly impacts on the circulating metabolome irrespective of COVID-19 status. (**A**) Overview of the study design: participants were grouped according to COVID-19 status and severity (negative, mild/asymptomatic, moderate, and severe), and according to HD status (HD and non-HD). Targeted metabolomics was performed from plasma samples, and metabolites were compared between groups. (**B**) Principal component analysis of targeted metabolomics separated non-HD from HD patients along PC2, and more advanced disease stages along PC1. (**C**) A schematic overview of contrasts between groups is depicted. Double arrows indicate the respective contrast, and integers indicate the number of deregulated metabolites.

**Figure 2 ijms-23-14089-f002:**
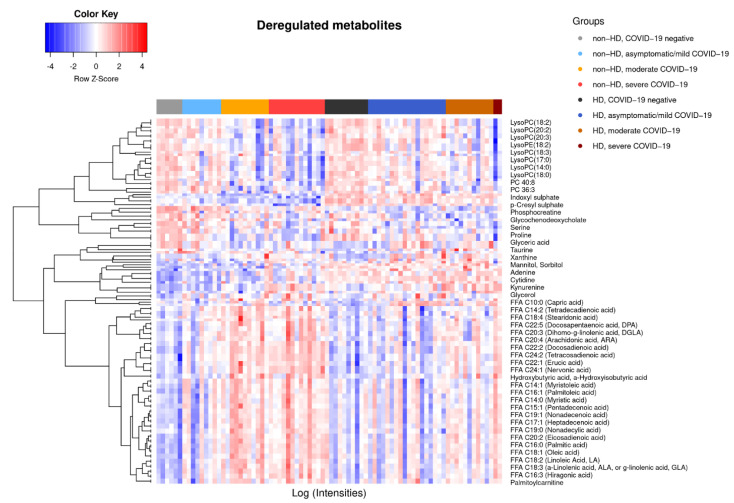
Significantly deregulated metabolites are shown in a heatmap as z-scores. Individuals are grouped according to HD and COVID-19 status.

**Figure 3 ijms-23-14089-f003:**
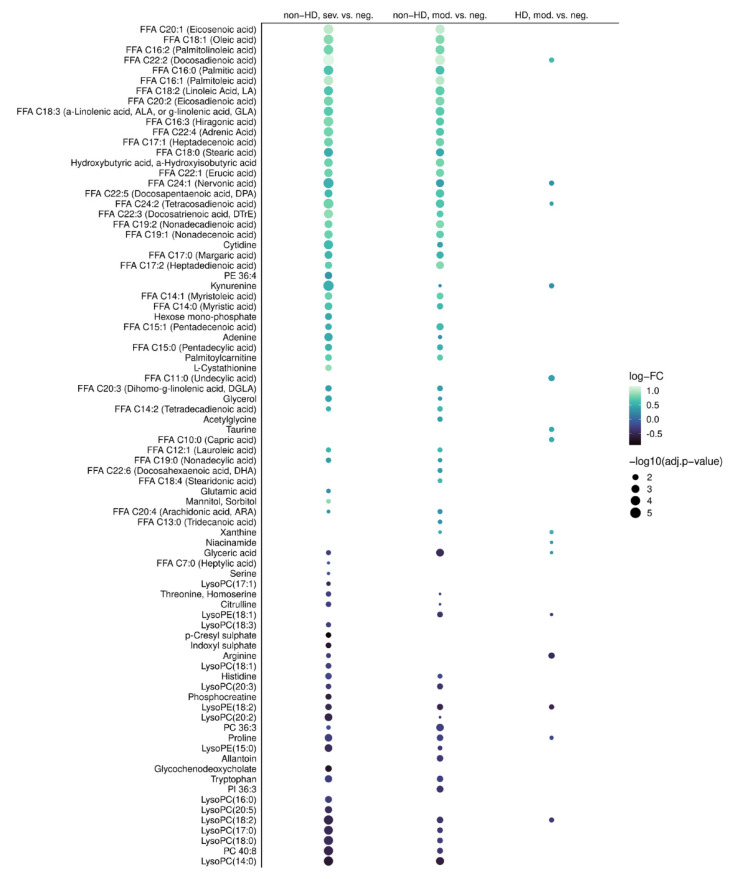
Metabolite deregulation in COVID-19 is less pronounced in HD. Significantly deregulated metabolites in the contrasts non-HD negative vs. non-HD moderate, non-HD negative vs. non-HD severe and HD negative vs. HD moderate are compared. Dot size indicates the adjusted *p*-value, the color scale represents the fold change of deregulated metabolites.

**Figure 4 ijms-23-14089-f004:**
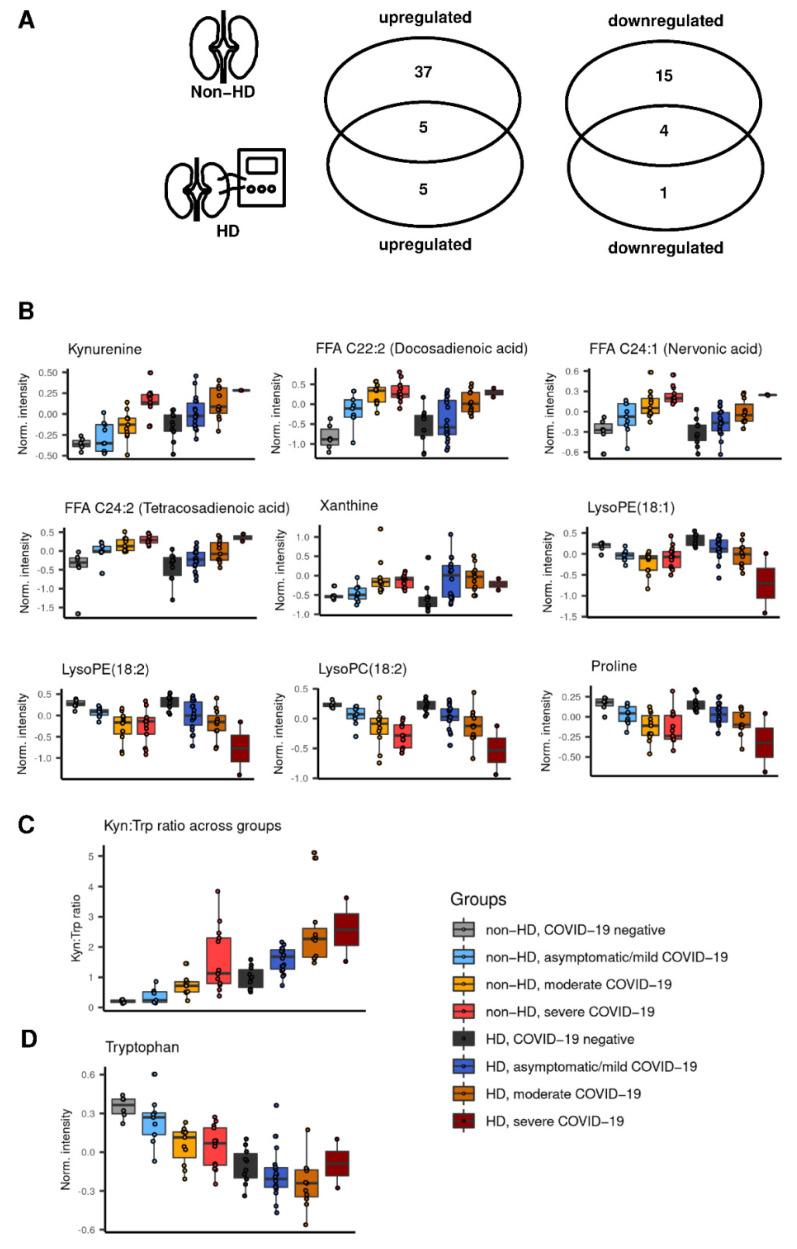
Overlapping metabolites in moderate COVID-19. (**A**) Venn diagrams depict overlapping and contrasting deregulated metabolites between non-HD and HD patients in the contrast negative vs. moderate COVID-19. (**B**) Box plots of metabolites that are jointly up- or downregulated in HD and non-HD with COVID-19 are shown. (**C**) The Kyn:Trp ratio across groups is shown. Lines mark the median, and boxes the interquartile range. Jittered dots represent individual values. (**D**) Trp levels in non-HD and HD patients with and without COVID-19 are shown as box plot.

**Figure 5 ijms-23-14089-f005:**
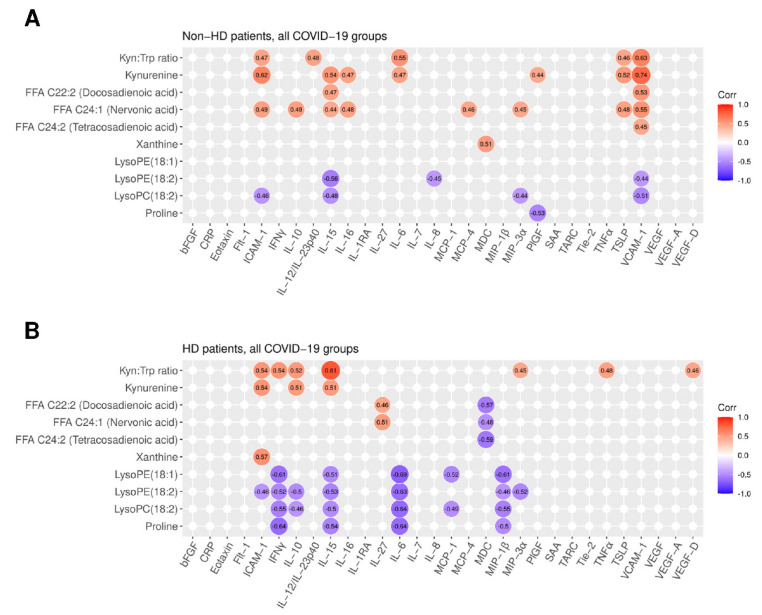
Correlation matrices of deregulated metabolites and cytokine/chemokine data. A graphical representation of significantly deregulated metabolites (COVID-19 negative vs. moderate) correlated with circulating cytokine and chemokine levels is shown for (**A**) non-HD and (**B**) HD patients. Spearman correlation is indicated by the blue-red color scale and ranges from −1 (blue) to 1(red). Only significant correlations (adjusted *p*-value ≤ 0.05) are displayed. *p*-values were adjusted with Benjamini–Hochberg method.

**Figure 6 ijms-23-14089-f006:**
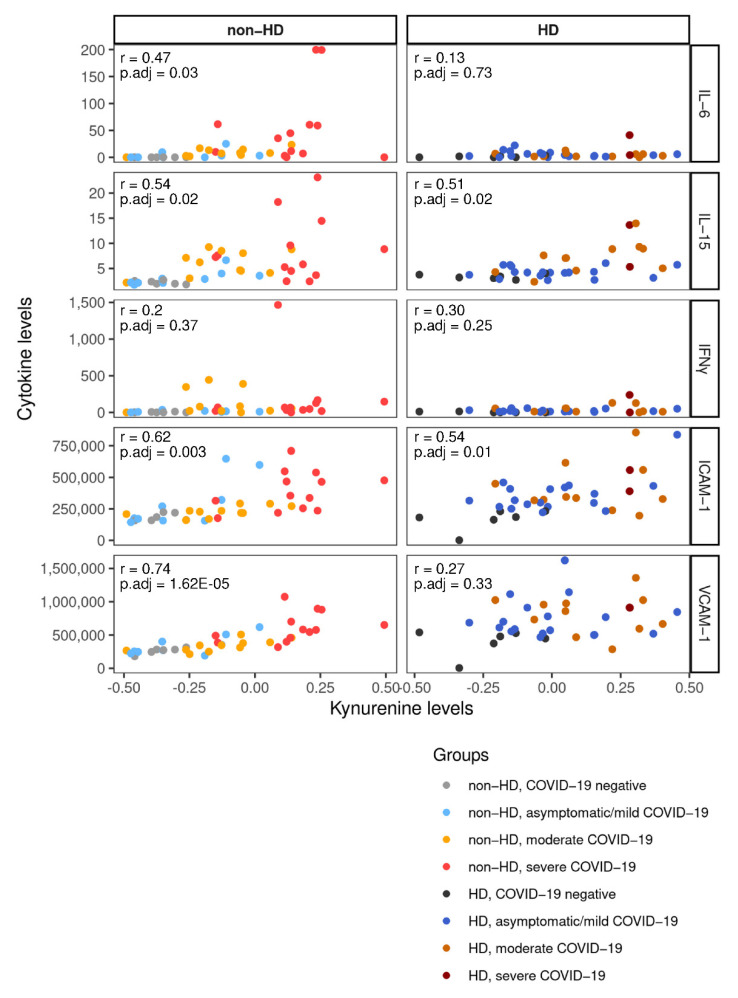
Scatter plots of Kyn (*x*-axis) with selected cytokines and chemokines (*y*-axis) in non-HD and HD patients are provided. Dots represent individual data points and dot colors reflect disease stages. Correlation coefficients (Pearson R) and adjusted *p*-values are given for each correlation.

**Table 1 ijms-23-14089-t001:** Top 10 metabolites and their relative contribution to principal components 1 and 2 for PC analysis plot (Figure 1B) are depicted.

PC 1		PC 2	
Metabolite	Contributions (%)	Metabolite	Contributions (%)
FFA C16:2 (Palmitolinoleic acid)	1.889	Cytidine	3.477
FFA C20:2 (Eicosadienoic acid)	1.879	Creatinine	3.316
FFA C17:1 (Heptadecenoic acid)	1.867	Gluconic acid	3.309
FFA C16:0 (Palmitic acid)	1.857	Glucuronic acid	3.191
FFA C19:1 (Nonadecenoic acid)	1.857	Orotidine	3.085
FFA C20:1 (Eicosenoic acid)	1.843	1-Methylhistidine	3.042
FFA C22:4 (Adrenic Acid)	1.841	ADMA	2.875
FFA C18:1 (Oleic acid)	1.818	Tryptophan	2.783
FFA C22:5 (Docosapentaenoic acid, DPA)	1.818	Trehalose	2.682
FFA C16:3 (Hiragonic acid)	1.807	Lactose	2.677

## Data Availability

Data have been deposited on MetaboLights—study number MTBLS6030.

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
