# Peer review of "Alterations in the Kynurenine–Tryptophan Pathway and Lipid Dysregulation Are Preserved Features of COVID-19 in Hemodialysis"

_ijms, 2022, doi:10.3390/ijms232214089_

Round 1
Reviewer 1 Report
In this study, the authors identified metabolic alterations induced by COVID-19, particularly in the Kyn-Trp pathway and in the lipid metabolism dysregulation.
The study is well-performed and well-written. I have read your manuscript carefully and have no suggestions to contribute as the methods and results are appropriate for the objective of the study. Furthermore, the discussion discusses all possible topics that the present study shows, including the limitations of the study.
I want to congratulate all the authors for their important work in understanding the molecular mechanisms underlying the severity of COVID-19.
Author Response
We thank the Reviewer for his/her generous comments on our manuscript!
Reviewer 2 Report
In the present study, the authors recruited 18 and nine asymptomatic/mild, 11 and 11 moderate, two and 13 severely affected, and 10 and six uninfected hemodialysis (HD) and non-HD patients to analyzed the plasma metabolites in different COVID-19 stages in patients with or without HD. The authors found Elevated kynurenine levels and lipid dysregulation, shown by an increase of circulating free fatty acids and a decrease of lysophospholipids, could distinguish patients with moderate COVID-19 from non-infected individuals in both groups. Kynurenine and lipid alterations were also associated with ICAM-1 and IL-15 levels in HD and non-HD patients.
1, The authors should provide Clinical characteristics and Baseline characteristics of patients. Results must be excluded if other factors are involved.
2. How did the authors divide COVID-19–positive patients into groups with asymptomatic/mild, moderate, and severely affected patients?
3. Does the metabolism correlate with renal function?
4. The relationship between tryptophan and COVID-19 IN HEMODIALYSIS is not well represented in the article.
Round 2
Reviewer 2 Report
This is an interesting manuscript and overall the study is well-conducted.